# Does attendance at a specialist antenatal clinic improve clinical outcomes in women with class III obesity compared with standard care? A retrospective case-note analysis

Fiona C Denison,[1] Heather MacGregor,[1] Laura I Stirrat,[1] Kerrie Stevenson,[1] Jane E Norman,[1] Rebecca M Reynolds[1,2]

► Prepublication history and additional material are available. To view these files please visit the journal online (http://dx.doi.org/10.1136/bmjopen-2016-015218)

[1]Tommy's Centre for Maternal and Fetal Health, MRC, University of Edinburgh Centre for Reproductive Health, University of Edinburgh, Edinburgh, UK
[2]British Heart Foundation Centre for Cardiovascular Research, University of Edinburgh, Edinburgh, UK

**Correspondence to**
Dr Fiona C Denison;
Fiona.Denison@ed.ac.uk

## ABSTRACT

**Objectives** To determine whether attendance at a specialised multidisciplinary antenatal clinic for women with class III obesity (BMI >40 kg/m$^2$) is associated with improved clinical outcomes compared with standard antenatal care.

**Design** Retrospective cohort study using routinely collected data from electronic patient record.

**Setting** Community and hospital based antenatal care.

**Participants** Women with a singleton pregnancy with class III obesity booked for antenatal care and delivered in one of two hospitals in NHS Lothian, Scotland, UK between 2008 and 2014. Maternal and offspring outcomes were compared in women who attended a specialised obesity clinic (n=511) compared with standard antenatal care (n=502).

**Main outcome measures** Included stillbirth, low birth weight, gestational diabetes, induction of labour and caesarean section.

**Results** Compared with standard care, women receiving specialist care were less likely to have a stillbirth (OR 0.12, 95% CI 0.06 to 0.97) and a low birthweight baby (OR 0.57, 95% CI 0.33 to 0.99) and more likely to be screened for (100% vs 73.6%; p<0.001) and diagnosed with (26.0% vs 12.5%; p<0.001) gestational diabetes, to require induction of labour (38.4% vs 29.9%; p=0.009), an elective (20.3% vs 17.7%; p<0.001) and emergency (23.9% vs 20.3%; p<0.001) caesarean section and attend antenatal triage one or more times during pregnancy (77.7% vs 53.1%; p<0.001). Women attending the specialist clinic had a higher BMI (44.5 kg/m$^2$ (4.3) vs 43.2 kg/m$^2$ (3.1); p<0.001) and were more likely to be nulliparous (46.0% vs 24.9%; p<0.001). There were no other differences in maternal demographic or maternal and offspring outcomes between groups.

**Conclusions** Attendance at a specialised antenatal clinic for obesity is associated with reduced rates of stillbirth and low birth weight and improved detection of gestational diabetes. The improvement in clinical outcomes is associated with an increase in healthcare attendance to obstetric triage and clinical interventions including induction of labour and caesarean section.

## Strengths and limitations of this study

► This study compares maternal and offspring outcomes in women with class III obesity who attend a specialist obesity antenatal clinic compared with those who received standard care.
► A strength of our study is that we were able to compare important clinical outcomes in women and offspring such as stillbirth and low birth weight.
► The use of routinely collected clinical data means that our results are relevant to clinical practice in which multiple different care pathways exist.
► The stillbirth findings and causality need to be interpreted with caution due to the small sample size and attenuation of findings in adjusted analyses.
► As a retrospective cohort study using routinely collected data from electronic patient record, results must be interpreted with caution because of potential bias from confounding factors.

20% of pregnant women are obese and 2% have very severe obesity (class III obesity, body mass index (BMI) ≥40 kg/m$^2$).[1] Maternal obesity is associated with increased risks for adverse maternal and offspring health including gestational diabetes mellitus (GDM), thromboembolic and hypertensive complications, caesarean section, macrosomia and stillbirth.[2–5] Managing these complications has significant cost implications for delivery of antenatal care.[2 4 6]

There is recognition that obstetric management of the obese should be consultant led and involve a multidisciplinary team to improve outcome.[7 8] These recommendations are embedded in clinical guidelines and standards of care produced by a number of countries.[8–13] However, there is a paucity of evidence demonstrating that multidisciplinary care and adherence to guidelines results in

## INTRODUCTION

Maternal obesity is the most common comorbidity of pregnancy. In the UK, approximately

improved maternal and offspring outcomes in maternal obesity. There is also less consensus about how multidisciplinary care should be delivered, and a concern that in areas of high obesity prevalence specialist obesity clinics are unlikely to be feasible due to cost and the numbers of women who would potentially need to be seen.[13]

Women with class III obesity are at a particularly high risk of adverse maternal and offspring outcome.[14] In 2008 we therefore set up a specialist antenatal clinic for women with class III obesity living in Edinburgh and the surrounding Lothian area with the aim of improving maternal and offspring outcomes. At their first antenatal appointment, which is generally prior to 12 weeks gestation, women with a BMI >40 kg/m$^2$ are offered referral to the specialist clinic or can choose to continue to receive standard antenatal care. We have a pan-Lothian guideline for clinical management of pregnancies in women with obesity (classes I, II and III) so that the same care pathway is offered, regardless of who or where it is delivered. All women with class III obesity should therefore receive the same standard of care. We hypothesised that maternal and offspring outcomes would be better in women who had their antenatal care provided by a multidisciplinary specialist clinic as opposed to receiving standard antenatal care. To test this hypothesis, we undertook a retrospective case-note review of all women with a BMI >40 kg/m$^2$ who delivered in Lothian between 2008 and 2014 and compared clinical outcomes in women who attended specialist antenatal care compared with those who received standard antenatal care.

## METHODS

### Study population

We performed a retrospective case-note review of all women with class III obesity with a singleton pregnancy who booked for antenatal care and delivered in either of two hospitals in the NHS Lothian trust between 2008 and 2014. The Simpson Centre for Reproductive Health at the Royal Infirmary of Edinburgh is a tertiary referral centre with more than 6500 deliveries per annum. St John's Hospital, Livingston, is a district general hospital with approximately 2600 deliveries per annum. Women were excluded if they had not delivered by the end of December 2014, had a multiple pregnancy (n=28) or booked later than 20 weeks gestation (n=18) because this meant they would have missed the gestational window for early screening for GDM.[15]

### Clinical care pathway

Women attending the specialist clinic at the Simpson Centre for Reproductive Health, Royal Infirmary of Edinburgh receive multidisciplinary consultant-led care throughout pregnancy from obstetricians, specialist midwives, diabetologists, anaesthetists, dieticians and other specialists as clinically indicated. At their first appointment (~10–16 weeks gestation), women are reviewed individually by a dietician with specialist

expertise in weight management during pregnancy and given tailored advice about healthy eating and weight management during pregnancy. They are advised to have early screening for GDM with a fasting blood glucose between 12 weeks and 16 weeks and late screening using a 75 g oral glucose tolerance test between 24 weeks and 28 weeks, as per the Scottish Intercollegiate Guidelines.[15] If a woman has pre-existing type 2 diabetes or is diagnosed with GDM during pregnancy, her care remains within the specialist clinic. At each visit, women are weighed, counselled about the maternal and offspring risks associated with maternal obesity, and their blood pressure is measured with appropriate sized cuffs. Women are commenced on 75 mg aspirin if they have additional risk factors for pre-eclampsia such as a blood pressure of >140/90 mm Hg at antenatal booking or primiparity as per national guidelines.[16] All women have postnatal thromboprophylaxis with low molecular weight heparin, with antenatal thromboprophylaxis being commenced if additional risk factors develop.[16] Fetal growth is monitored by serial growth scans at 28 weeks, 32 weeks and 36 weeks. All women receive a personalised delivery plan and an anaesthetic review in the third trimester to discuss intrapartum pain management with specific consideration given to obesity related comorbidities with implications for analgesia and anaesthesia.

Women who do not attend the specialist clinic receive guideline-based consultant-led care in hospital (tertiary or district general) or community-based antenatal clinics. The main difference between specialist and standard care is that if a woman receiving standard care develops a complication she needs to attend an additional separate specialist clinic, for example, a diabetes clinic in the event she develops gestational diabetes. For women who attend the obesity clinic, this care is centralised in a single multi-professional clinic.

To compare maternal and offspring outcomes by antenatal care setting, women were categorised as 'Specialist care' if they attended for two or more appointments at the specialist clinic with the first appointment being before 20 weeks. The rationale for this was that such women would have received early dietary advice and counselling about the importance of attending for early screening for GDM. Women who did not attend the specialist clinic were categorised as receiving 'standard care'.

### Data collection

Maternal and offspring data were acquired from the maternity electronic patient records database TRAK (supplied by Intersystems), clinical biochemistry database APEX (ApexHealthware) and the neonatal unit electronic patient records database BadgerNet (supplied by Clevermed) systems, with data being reported as per the RECORD checklist for reporting of observational studies using routinely collected health data (Supplementary table 1).

The following data were collected from the maternal record at booking: maternal age, BMI (kg/m$^2$), ethnicity

(white, other), parity (P0, P1, P2 or more), smoking status (current, former, never), deprivation quintile (a postcode-based Scottish Index of Multiple Deprivation from 2012 with five groups ranging from most deprived index (1) to least deprived index (5) [17] and systolic and diastolic blood pressures (mm Hg).

Maternal outcomes collected were hypertension (pre-existing, gestational, pre-eclampsia), diabetes (pre-existing, GDM), onset of labour (no labour, spontaneous onset, induced), delivery method (elective caesarean, emergency caesarean, instrumental, spontaneous vaginal), blood loss at delivery and antenatal obstetric triage attendances. The prevalence of GDM was determined according to (1) the rates of GDM from diagnoses entered into the electronic patient record and (2) evaluating whether blood glucose values found on the electronic databases conferred a diagnosis of GDM. Diagnostic accuracy of GDM was determined according to Scottish Intercollegiate Guidelines which used the WHO recommended thresholds[18] until March 2010 when updated thresholds were published based on the International Association of the Diabetes and Pregnancy Study Groups.[19]

Offspring outcomes collected were gender, birth weight, birthweight centile,[20] macrosomia (defined as birth weight ≥4000g), low birth weight (defined as birth weight ≤2500g), gestation of delivery, preterm birth (defined as birth <259 days gestation) and outcome (live birth, stillbirth).

All data were anonymised with personal identifiers removed before analysis. To maximise accuracy and to minimise missing data all records were reviewed by HM and LS, glucose data were reviewed by KS and LS with any discrepancies reviewed by FD, RR. For stillbirths, a perinatal pathologist examined placental pathology as is routine clinical practice. HM and LS independently identified risk factors and categorised the likely causality of the stillbirths. Stillbirth causation was checked and verified by a third investigator (FD). All investigators were blinded to whether a woman received 'specialist' or 'standard' care until risk factors and likely causality were agreed for all stillbirths.

## Statistical analysis

Data were analysed using Statistical Package for the Social Sciences (SPSS) V.21. Differences in the characteristics and clinical outcomes between the women who attended the specialist obesity clinic and those who received standard care were tested using Student's t-test if the variable was continuous or the $\chi^2$ test for categorical variables. Logistic regression was used to adjust for BMI and parity. A p value <0.05 was considered statistically significant.

## RESULTS
### Demographics

Maternal demographics are demonstrated in table 1. Compared with standard care women who attended the specialist clinic had a higher BMI, and were more likely to

**Table 1** Demographics of population

| | Specialist (n=511) | Standard (n=502) | p Value |
|---|---|---|---|
| Age (years; mean (SD)) | 29.8 (5.4) | 29.3 (5.5) | 0.11 |
| BMI (kg/m²; mean (SD)) | 44.5 (4.3) | 43.2 (3.1) | <0.001 |
| Ethnicity (n (%))* | | | 0.35 |
| White | 441 (94.6) | 432 (92.9) | |
| Other | 25 (5.4) | 33 (7.1) | |
| Parity (n (%)) | | | <0.001 |
| 0 | 235 (46.0) | 125 (24.9) | |
| 1 | 161 (31.5) | 212 (42.2) | |
| 2 or more | 115 (22.5) | 165 (32.9) | |
| Smoking status (n (%))* | | | 0.51 |
| Current | 45 (17.2) | 42 (13.7) | |
| Former | 63 (24.0) | 79 (25.7) | |
| Never | 154 (58.8) | 186 (60.6) | |
| Deprivation quintile (n (%))†* | | | 0.07 |
| 1 | 140 (27.7) | 108 (22.2) | |
| 2 | 141 (27.9) | 150 (30.9) | |
| 3 | 95 (18.8) | 107 (22.0) | |
| 4 | 66 (13.1) | 74 (15.2) | |
| 5 | 63 (12.5) | 47 (9.7) | |
| Systolic blood pressure (mm Hg; mean (SD)) | 122 (11.9) | 122 (11.1) | 0.79 |
| Diastolic blood pressure (mm Hg)† | 75 (9.0) | 75 (8.0) | 0.98 |

SD, standard deviation; BMI, body mass index.
*Missing data includes n=82 (8%) from ethnicity, n=444 (44%) from smoking and n=12 (1.2%) from deprivation quintile. Missing data are high from smoking as this was not a mandatory field on the electronic record until 2012.
†Deprivation quintile where 1 is the most and 5 the least deprived.

be primiparous. There were no differences in age, ethnicity, smoking status, systolic or diastolic blood pressures at booking between attenders and non-attenders. There was a trend towards deprivation levels being different in those attending for specialist compared with standard care with more women from both the least and most deprived attending specialist care.

### Maternal outcomes

Maternal outcomes are demonstrated in table 2. After excluding women with pre-existing type 1 and type 2 diabetes, all women who attended the specialist clinic had a screening test with sufficient information being collected to confirm or exclude a diagnosis of GDM. In contrast, 26.4% (128/484) of those receiving standard care either had no screening test for GDM or insufficient information was collected for a diagnosis of GDM to be made. The clinical diagnosis of GDM from the patient record matched the diagnosis from blood glucose levels in all women who attended the specialist clinic. In contrast, in

**Table 2** Maternal outcomes

| | Specialist n=511 | Standard n=502 | p Value |
|---|---|---|---|
| **Pre-existing comorbidities** | | | |
| Type 2 diabetes, n (%) | 2 (0.4) | 12 (2.4) | 0.008 |
| **Hypertensive complications** | | | 0.27 |
| Chronic hypertension, n (%) | 16 (1.6) | 11 (1.1) | |
| Gestational hypertension, n (%) | 18 (1.8) | 16 (1.6) | |
| Pre-eclampsia, n (%) | 31 (3.1) | 25 (2.5) | |
| **Gestational diabetes*** | | | |
| Screening/diagnostic test performed, n (%) | 496 (100) | 356 (73.6) | <0.001 |
| Prevalence, n (%) | 129 (26.0) | 61 (12.5) | <0.001 |
| **Labour and delivery** | | | |
| Onset labour, n (%) | | | 0.009 |
| No labour | 111 (21.7) | 109 (21.7) | |
| Spontaneous onset | 204 (39.9) | 243 (48.4) | |
| Induction | 196 (38.4) | 150 (29.9) | |
| Delivery method, n (%) | | | |
| Elective caesarean | 103 (20.2) | 89 (17.7) | <0.001 |
| Emergency caesarean | 122 (23.9) | 102 (20.3) | |
| Instrumental | 56 (11.0) | 23 (4.6) | |
| Spontaneous vertex | 229 (44.9) | 288 (57.4) | |
| Blood loss at delivery (mL; mean (SD)) | 575 (464) | 465 (387) | <0.001 |
| Obstetric triage attendances, n(%) | | | <0.001 |
| 0 | 108 (21.1) | 229 (45.6) | |
| 1 | 132 (25.8) | 104 (20.7) | |
| 2 | 93 (18.2) | 70 (13.9) | |
| 3 or more | 172 (33.7) | 93 (18.5) | |

*Denominator excludes women with pre-existing diabetes (type 1 or 2) or those who were not managed at the tertiary referral centre. In women who attended for specialist and standard care, the prevalence is based on blood glucose levels and not the clinical diagnosis recorded in the notes.

those receiving standard care, when the notes and actual blood glucose values were compared, the 'wrong' diagnosis was made in 17 women. One woman was incorrectly diagnosed with GDM when her screening test for GDM was normal. A further 16 woman had a positive diagnostic test for GDM according to glucose values obtained during a glucose tolerance test but the diagnosis was missed and these women were incorrectly labelled as not having GDM (and did not therefore receive treatment).

Compared with those receiving standard care, women who attended the specialist clinic were more likely to have their labour induced, to have a caesarean or instrumental vaginal delivery. Specialist clinic attenders had a higher blood loss at delivery than those receiving

standard care even after adjusting for mode of delivery, BMI, age and parity (p=0.02). They were also more likely to attend obstetric triage one or more times during pregnancy. Rates of pre-existing chronic hypertension and hypertensive complications (gestational hypertension and pre-eclampsia) were low in both attenders and non-attenders. Rates of type 2 diabetes were higher in non-attenders compared with attenders.

### Offspring outcomes

The clinical details for the offspring outcomes are demonstrated in table 3. Compared with standard care, women attending specialist care were less likely to have a stillbirth (OR 0.12, 95% CI 0.06 to 0.97) and a low birth-weight baby (OR 0.57, 95% CI 0.33 to 0.99). The lower stillbirth outcomes in women who attended specialist care were attenuated in analyses adjusting for BMI and parity (adjusted OR (AOR) 0.14, 95% CI 0.02 to 1.17) but the lower risk of having a low birthweight baby was strengthened in adjusted analyses (AOR 0.52, 95% CI 0.29 to 0.93). The clinical details of the women who had a stillbirth are demonstrated in table 4. In women attending for standard care, an additional risk factor for stillbirth was identified in seven women and a probable cause for stillbirth was identified in all eight women. No additional risk factors or cause were identified in the one woman who had a stillbirth who attended the specialist clinic.

### DISCUSSION

In this retrospective case-note review, we demonstrated that women with class III obesity who attended a specialist multidisciplinary antenatal clinic were less likely to have a stillbirth and low birthweight infant and more likely to be tested, correctly diagnosed with and treated for GDM, and to have an induction of labour, caesarean section and higher blood loss at delivery compared with those receiving standard antenatal care. These differences in outcomes were accompanied by increased attendance at obstetric triage.

### Main findings

A key study finding was that rates of stillbirth and low birth weight were lower in women who attended the clinic compared with those who did. Compared with standard care, women who attended the specialist clinic had a higher BMI, and were more likely to be primiparous. Given that primiparity and higher BMI are independently associated with increased risk of stillbirth and low birth weight,[21–23] we expected that rates of stillbirth and low birth weight would be higher in women receiving specialist as compared with standard care. However, we found the converse to be the case, with fewer stillborn and low birthweight babies being born to women attending the specialist clinic, even after adjusting for parity and BMI. We are uncertain why rates of low birth weight are lower in women attending the specialist clinic since there

**Table 3** Offspring outcomes

|  | Specialist n=511 | Standard n=502 | Significance (p value) |
|---|---|---|---|
| Gender, n (%) |  |  | 0.34 |
| Female | 238 (46.6) | 249 (49.6) |  |
| Male | 273 (53.4) | 253 (50.4) |  |
| Birth weight (g; mean (SD)) | 3576 (635) | 3559 (664) | 0.69 |
| Macrosomia,* n (%) | 31 (6.1) | 26 (5.2) | 0.54 |
| Low birth weight,† n (%) | 21 (4.1) | 35 (7.0) | 0.04 |
| Gestation (days; mean (SD)) | 277 (14.1) | 277 (14.7) | 0.82 |
| Preterm birth,‡ n (%) | 40 (7.8) | 39 (8.4) | 0.97 |
| Outcome, n (%) |  |  |  |
| Live birth | 510 (99.8) | 494 (98.4) | 0.02 |
| Stillbirth | 1 (0.2) | 8 (1.6) |  |

*Macrosomia defined as birth weight of 4000 g or more.
†Low birth weight defined as birth weight of 2500 g or lower.
‡Preterm birth defined as birth before 259 days gestation.

are no differences in the length of gestation or frequency of preterm birth.

The stillbirth rate in women who attended the specialist clinic was 2 per 1000 compared with a rate of 7 per 1000 for women with a BMI >40 kg/m$^2$ who delivered in Scotland in 2011–2012.[24] To validate this finding, three investigators who were blinded to whether women received specialist or standard care independently checked the stillbirth data. It was striking that additional risk factors were identified in seven and a cause for stillbirth identified in all eight women who received standard care and who had a stillbirth but no additional risk factors or cause were identified in the one woman who had a stillbirth who attended the specialist clinic. We accept that rates of unexplained stillbirth are generally reported as being 20%–25% which is much higher than what we found in our study. We therefore acknowledge that the stillbirth findings and causality need to be interpreted with caution

**Table 4** Details of stillbirths

| Case | Demographics Age (years) | Parity | BMI (kg/m$^2$) | Risk factors | Outcome | Birthweight centile* | Cause |
|---|---|---|---|---|---|---|---|
| ST1 | 31 | P2 | 42 | Smoker, type 2 diabetes, RFM | 33+5 weeks, boy, 2050 g | 25th–50th | Uncontrolled hypertension, abruption |
| ST2 | 32 | P1 | 42 | No risk factors | 30+5 weeks, girl, 700 g | <3rd | IUGR, placental insufficiency |
| ST3 | 38 | P4 | 42 | RFM | 37 weeks, boy, 2720 g | 10th–25th | Severe pre-eclampsia, abruption |
| ST4 | 32 | P2 | 45 | Smoker, RFM | 36 weeks, boy, 2160 g | 5th–10th | Acute intrauterine hypoxia |
| ST5 | 26 | P2 | 47 | Smoker, RFM, isolated congenital anomaly | 35+5 weeks, girl, 2155 g | 10th–25th | Congenital anomaly |
| ST6 | 32 | P2 | 52 | Smoker | 30+5 weeks, girl, 1620 g | 75th–90th | Abruption |
| ST7 | 27 | P2 | 40 | Type 2 diabetes, RFM | 38+2 weeks, boy, 3370 g | 50th–75th | Poorly controlled diabetes |
| ST8 | 21 | P0 | 40 | Smoker | 26+3 weeks, girl, 750 g | 25th–50th | IUGR, placental insufficiency |
| SP1 | 20 | P1 | 41 | No risk factors | 39+5 weeks, boy, 3725 g | 50th–75th | Unexplained |

*Birth weight centile defined by Bonellie et al.
BMI, body mass index; RFM, reduced fetal movements; IUGR, intrauterine growth restriction; ST, standard; SP, specialist.

due to the small sample size and attenuation of findings in adjusted analyses. However it is tempting to speculate that the continuity of care together with the education of women by the multidisciplinary clinic team raised increased awareness of the importance of risk factors such as reduced fetal movements and this may have led to them presenting earlier to obstetric triage and being induced prior to stillbirth occurring. Future studies such as the AFFIRM clinical trial (NCTT01777022, due to complete in 2017) are designed to address this in the general antenatal population.

## Strengths and limitations

A strength of our study is that we were able to compare important clinical outcomes in women and offspring such as stillbirth. We also used routinely collected clinical data meaning that our results are relevant to clinical practice in which multiple different care pathways exist. We accept that a limitation of our study is that this was a retrospective case-note review and our sample size was therefore limited by the study population. For the majority of data fields, other than smoking status (43.8% missing), there was a relatively low proportion of missing data. For the smoking variable, this was due to smoking status not being a mandatory field for recording on the electronic clinical record prior to 2012. The study was also not randomised, so women could choose whether to attend the specialist clinic. However, apart from differences in maternal BMI (although a small difference of uncertain clinical significance) and primiparity and a trend towards differences in deprivation status between women who attended specialist compared with standard care, all other demographic factors were comparable between groups. Given that the clinical outcomes were better in women attending the specialist clinic who were arguably at higher risk than those attending standard care due to their higher BMI and more likely to be primiparous, we believe that our finding that multidisciplinary care improves clinical outcomes in pregnant women with class III obesity compared with standard care is clinically important.

## Interpretation

Pregnancy outcomes tend to be worse in women who either do not attend or under-attend any antenatal care, regardless of whether their pregnancies are categorised as low-risk or high-risk.[25] However, although we categorised women into women who attended specialist and standard care this was only in relation to how their antenatal care was organised and not whether they did or did not attend any antenatal care. In 1993, the landmark Changing Childbirth Report,[26] which was built on the 1992 Winterton Report, reversed the official policy that hospital is always the safest place for birth and emphasised the importance of maternal choice, control and continuity of carer for women. These recommendations, which were made over 20 years ago are still as relevant today, and frame the rhetoric and delivery of antenatal care across the UK.[27–30] In Lothian, all women receiving community-led care have a named midwife who coordinates their care. This midwife is part of a community team which has a defined case-load. This model ensures that there is continuity of care for a woman at both the individual midwife and midwifery team level. If a woman is categorised as having a high risk pregnancy (such as would be the case in women with class III obesity), she is also designated a named consultant to oversee her care. Despite this model of continuity of care, our study demonstrates that maternal and offspring outcomes are better in women who attend a hospital-based specialist clinic compared with those who receive standard antenatal care.

Although specialist clinics have been advocated as a way of improving maternal and offspring outcomes, there is currently a paucity of evidence from randomised controlled trials about the benefits and harms of specialist antenatal clinics compared with standard antenatal care for women.[31] For example, systematic reviews of randomised controlled trials have concluded that there is currently limited information to assess the role of specialist antenatal clinics for women with a multiple pregnancy[32] and no clear evidence that specialist clinics reduce the number of preterm births.[33] Given that the antenatal care pathway followed was the same in women who attended the specialist clinic and those who received standard care, it is not clear why maternal and offspring outcomes were better in women who attended the specialist clinic. A recent systematic review by Sandall *et al* highlighted the importance of continuity of care, demonstrating that pregnant women receiving midwife-led continuity models of care had at least comparable clinical outcomes and were likely to experience less intervention.[34] It is therefore plausible that the continuity of care that the specialist multidisciplinary team provided enabled compromised pregnancies to be identified more accurately and interventions such as induction of labour to be targeted more appropriately compared with those women receiving standard care. It is also possible that staff providing standard antenatal care have less experience of class III obesity and poorer access to appropriate facilities and equipment which may have adversely impacted their ability to provide optimal antenatal care to these high-risk women.

## CONCLUSION

In summary, our study demonstrates that attendance at a multidisciplinary specialist antenatal clinic improves maternal and offspring outcomes in women with class III obesity. This challenges current recommendations that women with very class III obesity can be effectively managed outside a specialist service. Further research is needed to identify the most appropriate and economic model of care for women with class III obesity to optimise maternal and offspring outcomes.

**Acknowledgements** The authors thank Mr Allyn Dick for assistance in extracting the clinical data from maternity TRAK.

**Contributors** FD and RR conceived the study and drafted the paper. FD, HM, LS, KS, JN and RR designed the study. FD, HM, LS, KS and RR acquired and analysed the data. All authors interpreted the data, revised the paper critically for important intellectual content and approved the final version.

**Funding** Tommy's the Baby Charity. The work was carried out in the MRC Center for Reproductive Health: Center Grant MRC MR/N022556/1.

**Competing interests**  None declared.

**Ethics approval**  Local audit committee.

**Provenance and peer review**  Not commissioned; externally peer reviewed.

**Data sharing statement**  Anonymised data will be available for data sharing subject to Caldicott approvals.

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
