## [Reviewer comments · BMJ Open]

ARTICLE DETAILS

TITLE (PROVISIONAL)	Does attendance at a specialist antenatal clinic improve clinical outcomes in women with Class III obesity compared to standard care? : a retrospective case-note analysis.
AUTHORS	Denison, Fiona; MacGregor, Heather; Stirrat, Laura; Stevenson, Kerrie; Norman, Jane; Reynolds, Rebecca

VERSION 1 - REVIEW

REVIEWER	Tomasina Stacey University of Leeds UK
REVIEW RETURNED	09-Dec-2016

GENERAL COMMENTS	This is an interesting and relevant paper, especially in the light of the growing rate of severe obesity. The paper is well written and well presented. It appears that the comparison being made is not just between a specialist clinic and a general clinic, but between a clinic in a tertiary unit and one in a district general hospital. This needs clarification and if it is the case, then the potential impact of the different levels of expertise and experience of the staff and facilities needs to be considered and included in the discussion and interpretation of the results. Is there some explanation for the high % of missing data for smoking status- a key factor in the outcomes of interest? The authors make a comment in the introduction regarding the potential cost implications of specialist clinics, but there is no health economic assessment included in the paper. Some form of cost analysis would greatly enhance the usefulness and potential impact of the findings from this study.
---

REVIEWER	Melissa Rosentein, MD MAS University of California, San Francisco USA
REVIEW RETURNED	21-Dec-2016

GENERAL COMMENTS	This is an excellent manuscript – a great study and very well-written. They are thoughtful and circumspect in describing their results, which include a remarkable decrease in stillbirths among women attending the specialist clinic. A few comments:
---

	For ease of reading the tables, would change the categories from “Attender” and “Non-attender” (even though this was explained in the text, it still does sound like the “non-attenders” were less compliant with antenatal care). Maybe “Specialist” and “Standard” or something like that? I know it is not technically “statistically significant” (p value 0.07) but I think it would be OK to comment that the deprivation levels among the specialist and standard groups are a bit different, with more deprived women in the specialist clinic – further bolstering the claim that the clinic itself is helpful. At the same time, while the p value for the difference in BMI is very significant, there is only a 1 point difference between the groups, so I’m not sure that is terribly clinically significant. I think the triage attendance differences is also striking. My guess is that this and the increased use of induction is what led to lower rates of stillbirth. One question is whether the rates of low birthweight were due to differences in gestational ages? That doesn’t seem to be the case, since there were no differences in the length of gestation or the frequency of preterm birth, but maybe this could be pointed out explicitly in the text. It is hard to explain that difference, since we don’t usually think of low birthweight as something that can be prevented, only identified inasmuch it is a risk factor for stillbirth. I think the outcomes of preeclampsia and hypertension should also be in Table 3, since 2 of the stillbirths were complicated by hypertension. There were 2 IUGR stillbirths in the standard care group – the authors mention that there is no real difference in the care recommendations – is there any data about the number of ultrasounds performed? How about modified BPPs or NSTs? I think the conclusions are sound and also cautious. They are as surprised as I am about the findings, and while they may be spurious, they deserve publication and further research.
--	--

VERSION 1 – AUTHOR RESPONSE

REVIEWER: 1

C1: This is an interesting and relevant paper, especially in the light of the growing rate of severe obesity. The paper is well written and well presented. It appears that the comparison being made is not just between a specialist clinic and a general clinic, but between a clinic in a tertiary unit and one in a district general hospital. This needs clarification and if it is the case, then the potential impact of the different levels of expertise and experience of the staff and facilities needs to be considered and included in the discussion and interpretation of the results.

R1: We thank the reviewer for their supportive comments. We have clarified the comparison made and commented on this in the discussion (lines 378 – 381). As suggested by reviewer 2, we have changed the terminology to specialist care and standard care.

C2: Is there some explanation for the high % of missing data for smoking status- a key factor in the outcomes of interest?

R2: We agree that smoking is a key factor in the outcomes of interest and that a high % of this data was missing. This variable was not a mandatory field earlier in the time period for data collection, thus this data was missing equally for both smokers and non-smokers.

C3: The authors make a comment in the introduction regarding the potential cost implications of specialist clinics, but there is no health economic assessment included in the paper. Some form of cost analysis would greatly enhance the usefulness and potential impact of the findings from this study.

R3: We agree that a health economic assessment would be very valuable but this was not the focus of the current paper. We agree that assessing this (in a future paper) would be very useful.

REVIEWER: 2

C1: This is an excellent manuscript – a great study and very well-written. They are thoughtful and circumspect in describing their results, which include a remarkable decrease in stillbirths among women attending the specialist clinic.

R1: We thank the reviewer for their supportive comments about our manuscript.

C2: For ease of reading the tables, would change the categories from “Attender” and “Non-attender” (even though this was explained in the text, it still does sound like the “non-attenders” were less compliant with antenatal care). Maybe “Specialist” and “Standard” or something like that?

R2: We agree that “specialist” and “standard” is clearer and have changed the categories from “attender” and “non-attender” to “specialist” and “standard” care throughout.

C3: I know it is not technically “statistically significant” (p value 0.07) but I think it would be OK to comment that the deprivation levels among the specialist and standard groups are a bit different, with more deprived women in the specialist clinic – further bolstering the claim that the clinic itself is helpful. At the same time, while the p value for the difference in BMI is very significant, there is only a 1 point difference between the groups, so I’m not sure that is terribly clinically significant.

R3: As requested, we have now commented on the differences in deprivation between women attending the “specialist” clinic and those attending “standard” care and acknowledged that the differences in BMI, although statistically significant, may not be clinically significant (lines 226 – 231).

C4: I think the triage attendance differences is also striking. My guess is that this and the increased use of induction is what led to lower rates of stillbirth.

R4: We agree with the reviewer and have added this to the discussion (lines 310 - 311).

C5: One question is whether the rates of low birthweight were due to differences in gestational ages? That doesn’t seem to be the case, since there were no differences in the length of gestation or the frequency of preterm birth, but maybe this could be pointed out explicitly in the text. It is hard to explain that difference, since we don’t usually think of low birthweight as something that can be prevented, only identified inasmuch it is a risk factor for stillbirth.

R5: We agree that the differences in rates of low birthweight are difficult to explain and, as requested

by the reviewer now explicitly state that this can not be explained by differences in length of gestation or frequency of preterm birth (lines 295 - 297).

C6: I think the outcomes of preeclampsia and hypertension should also be in Table 3, since 2 of the stillbirths were complicated by hypertension.

R6: The outcomes of chronic hypertension, gestational hypertension and pre-eclampsia are already included in Table 2 which describes the maternal outcomes in women.

C7: There were 2 IUGR stillbirths in the standard care group – the authors mention that there is no real difference in the care recommendations – is there any data about the number of ultrasounds performed? How about modified BPPs or NSTs?

R7: We do not have information about the number of ultrasounds, modified biophysical profiles or non-stress tests so unfortunately cannot comment on this point.

C8: I think the conclusions are sound and also cautious. They are as surprised as I am about the findings, and while they may be spurious, they deserve publication and further research.

R8: We thank the reviewer for their support.

VERSION 2 – REVIEW

REVIEWER	Tomasina Stacey University of Leeds UK
REVIEW RETURNED	30-Jan-2017

GENERAL COMMENTS	I feel that the authors have adequately addressed the minor comments from the original review.
--

REVIEWER	Melissa Rosentein, MD MAS University of California, San Francisco USA
REVIEW RETURNED	10-Feb-2017

GENERAL COMMENTS	This version addresses my prior concerns and I think is acceptable for publication in its current form.
---